# Does behavioral thermal tolerance predict distribution pattern and habitat use in two sympatric Neotropical frogs?

**Juan C. Díaz-Ricaurte**[1,2,3☉]*, **Filipe C. Serrano**[2☉], **Estefany Caroline Guevara-Molina**[4], **Cybele Araujo**[5], **Marcio Martins**[2‡]

1 Programa de Pós-Graduação em Ecologia Aplicada, Escola Superior de Agricultura Luiz de Queiroz, Piracicaba, São Paulo, Brazil, 2 Semillero de Investigación en Ecofisiología y Biogeografía de Vertebrados, Grupo de investigación en Biodiversidad y Desarrollo Amazónico (BYDA), Programa de Biología, Universidad de la Amazonia, Florencia, Caquetá, Colombia, 3 Departamento de Ecologia, Instituto de Biociências, Universidade de São Paulo, São Paulo, São Paulo, Brazil, 4 Departamento de Fisiologia, Instituto de Biociências, Universidade de São Paulo, São Paulo, São Paulo, Brazil, 5 Instituto Florestal, Seção de Animais Silvestres, Horto Florestal, São Paulo, São Paulo, Brazil

☉ These authors contributed equally to this work.
‡ These authors are senior author on this work.
* juan.diaz@usp.br

**Data Availability Statement:** All relevant data are within the manuscript and its Supporting Information files.

**Funding:** MM thanks Fundação de Amparo à Pesquisa do Estado de São Paulo for a grant

## Abstract

Environmental temperatures are a major constraint on ectotherm abundance, influencing their distribution and natural history. Comparing thermal tolerances with environmental temperatures is a simple way to estimate thermal constraints on species distributions. We investigate the potential effects of behavioral thermal tolerance (i. e. Voluntary Thermal Maximum, $VT_{Max}$) on anuran local (habitat) and regional distribution patterns and associated behavioral responses. We tested for differences in Voluntary Thermal Maximum ($VT_{Max}$) of two sympatric frog species of the genus *Physalaemus* in the Cerrado. We mapped the difference between $VT_{Max}$ and maximum daily temperature ($VT_{Max}$—$ET_{Max}$) and compared the abundance in open and non-open habitats for both species. *Physalaemus nattereri* had a significantly higher $VT_{Max}$ than *P. cuvieri*. For *P. nattereri*, the model including only period of day was chosen as the best to explain variation in the $VT_{Max}$ while for *P. cuvieri*, the null model was the best model. At the regional scale, $VTMax$—$ET_{Max}$ values were significantly different between species, with *P. nattereri* mostly found in localities with maximum temperatures below its $VT_{Max}$ and *P. cuvieri* showing the reverse pattern. Regarding habitat use, *P. cuvieri* was in general more abundant in open than in non-open habitats, whereas *P. nattereri* was similarly abundant in these habitats. This difference seems to reflect their distribution patterns: *P. cuvieri* is more abundant in open and warmer habitats and occurs mostly in warmer areas in relation to its $VT_{Max}$, whereas *P. nattereri* tends to be abundant in both open and non-open (and cooler) areas and occurs mostly in cooler areas regarding its $VT_{Max}$. Our study indicates that differences in behavioral thermal tolerance may be important in shaping local and regional distribution patterns. Furthermore, small-scale habitat use might reveal a link between behavioral thermal tolerance and natural history strategies.

(#2018/14091-8) and Conselho Nacional de Desenvolvimento Científico e Tecnológico for a research fellowship (# 306961/2015-6). JCDR and FCS thank Coordenação de Aperfeiçoamento de Pessoal de Nível Superior – Brasil (CAPES) – Finance Code 001. The funders had no role in study design, data collection and analysis, decision to publish, or preparation of the manuscript.

**Competing interests:** The authors have declared that no competing interests exist.

# Introduction

Environmental temperatures are a major constraint on ectotherm abundance and diversity, influencing their distribution and natural history [1–3]. Several studies have explored environmental constraints on ectothermic vertebrates at regional and global scales [1, 4]. The physiological performance of individuals can be negatively affected by high environmental temperatures [5], which can lead to declining populations and/or local extinctions [6, 7]. Thus, knowing species thermal tolerance and exploring how environmental temperatures might affect their physiology and restrict their distribution is of primary concern for long-term conservation, especially under current global warming crisis (e.g. [8, 9]), as well as habitat disturbance causing microclimate changes (e.g. habitat fragmentation; [10]).

However, thermal tolerances are rarely taken into account in studies that focus on local distribution and habitat use. For instance, many studies infer potential distribution of species using solely environmental temperatures from occurrence localities to model their niche [11–14]. While the broad geographical range of a species most likely reflects its thermal tolerance (e.g. [15, 16]), local factors might also play a role in shaping abundance and distribution. At a local scale, high environmental temperatures and its daily variation in the microhabitats of small ectotherms (e.g. anurans and lizards) impose physiological constraints on their activity patterns and habitat use [12]. For example, in habitats where direct sunlight is limited, the variation in temperatures is lower than in open habitats, suggesting a possible interplay between thermal tolerance and habitat use [17]. However, studies that relate how thermal tolerances affect habitat use and distribution are scarce.

Thermal tolerances can be behavioral, when an animal moves or adjusts its body posture to thermoregulate, or physiological if it does not move but uses other strategies such as increased respiration rates [3]. Behavioral and physiological thermal tolerances impact not only species ranges, but also the distribution and abundance patterns of their populations [3]. Identifying thermal tolerance thresholds (i.e. measurable thermal limits) outside the range of preferred body temperatures (PBT) for thermoregulation (see [18]) allows for the identification of temperatures that directly affect the behavioral and physiological thermal tolerance of ectothermic organisms. One of the thresholds related to PBT is the Voluntary Thermal Maximum ($VT_{Max}$), which represents a behavioral thermal tolerance measure. $VT_{Max}$ is the maximum temperature that an organism will endure before trying to move to a place with a lower temperature, thus trying to maintain its body temperature within its range of PBT [3, 18, 19]. If an individual fails to respond to its $VT_{Max}$, an increase in body temperature will expose it to its physiological thermal limit (i.e. its Critical Thermal Maximum), which can lead to functional collapse and consequently death due to overheating [19, 20]. Therefore, the behavioral response to upper limits might represent a more informative ecological threshold to identify thermal constraints on habitat use and geographic distribution [3, 8, 13]. Contrary to the Critical Thermal Maximum, the exposure to the $VT_{Max}$ does not induce an immediate loss of locomotion [3, 21]. Therefore, $VT_{Max}$ can more realistically portray changes in species behavior associated with their natural history.

Behavioral thermal tolerances can be influenced by factors such as reproductive status, sex, photoperiod, and hydration state [18, 22]. Additionally, thermal tolerances such as the $VT_{Max}$ might decrease with body size: due to thermal inertia, larger animals might have slower heating and cooling rates than small animals, which increases the exposed time to stressful thermal conditions [23, 24]. Thus, understanding the effects of these variables on the $VT_{Max}$ might help to evaluate its impact on habitat use and geographic distribution.

Herein we address the question: Does $VT_{Max}$ determine habitat use and regional distribution patterns in a pair of congeneric frogs, *Physalaemus cuvieri* and *P. nattereri*, which are

widely sympatric in the savannas of Central Brazil? Our hypothesis is that, for being a measure that reflects avoidance of stressful thermal conditions, $VT_{Max}$ determines both habitat use and geographic distribution in these species. If $VT_{Max}$ decreases with body size (see above; [23, 24]), we predict that $VT_{Max}$ is lower in the larger species (*P. nattereri*). Furthermore, if $VT_{Max}$ determines habitat use and geographic distribution, we predict that (i) the species with lower $VT_{Max}$ is less abundant in open habitats, with higher environmental temperatures, and that (ii), regarding geographic distribution, both species occur mostly in localities where the maximum environmental temperature is below their $VT_{Max}$. We expect that our results can contribute to assess the vulnerability of Neotropical frogs to climate change by integrating their behavioral thermal tolerances with their habitat use and distribution patterns, in order to identify areas with potential stressful climatic conditions to their populations.

## Materials and methods

### Focal species

Most species of the genus *Physalaemus* have sympatric populations along extensive areas, such as *Physalaemus nattereri* [25] and *Physalaemus cuvieri* [26] (see [27]), which are widespread in central South America [25, 26]. These species belong to different clades within *Physalaemus* (*P. signifer* and *P. cuvieri* clades, respectively; [28]). *Physalaemus nattereri* has a stout body, a moderate to large size (adult snout-to-vent length of 29.8–50.6 mm) and is endemic to the Cerrado, whereas *P. cuvieri* has a slenderer body, a smaller size (snout-to-vent length of adults 28–30 mm) and occurs throughout the Cerrado, in southern portions of the Amazon Forest and in the Atlantic Forest [29]. Although the populations traditionally assigned to *P. cuvieri* (see [27]) may include more than one cryptic species (see [28]), most of the distribution of *P. cuvieri* in the Cerrado correspond to a single lineage (Lineage 2 in [28]). These two species also differ in their biology. While *P. cuvieri* uses several aquatic habitats for reproduction and seeks shelter during the day in previously-dug burrows, *P. nattereri* breeds mostly in temporary puddles and buries itself in the soil during the day aided by metatarsal tubercles (S1 Fig) on its hind feet [29–31].

### Physiological parameters

**Capture and maintenance of individuals.**   Fieldwork was carried out at Estação Ecológica de Santa Bárbara (22˚49'2.43"S, 49˚14'11.29"W; WGS84, 590 m elevation), one of the few remnants of Cerrado savannas in the state of São Paulo, Brazil, with a total area of 2,712 ha [32]. The climate is Humid subtropical [33], with temperatures averaging 24˚C and 16˚C during January and July, the hottest and coldest months, respectively. The average annual rainfall is 1100–1300 mm, with marked dry and wet seasons (approximately April to September and October to March, respectively; [32]). The landscape not only consists of open grassland and savanna-type formations, such as 'campo sujo' and 'campo cerrado', but also of non-open vegetation types such as 'cerrado *strictu sensu*' (dense savanna) and 'cerradão' (cerrado woodland). Between 24 and 28 September 2018, we captured 14 individuals of *P. nattereri* and 20 of *P. cuvieri* in pitfall traps with drift fences [34, 35] and these individuals were housed individually in plastic boxes at room temperature. This study was conducted under a permit by Comissão de Ética no Uso de Animais (CEUA #2325141019) of Instituto Butantan. All animals were alive after the experiments described below and were released the following morning at the site of capture.

**Measurements of the Voluntary Thermal Maximum ($VT_{Max}$).**   To obtain the $VT_{Max}$ for each species, we measured each individual at 100% hydration level less than 24 hours after capture. To reach maximum hydration level, each individual was placed in a cup with water *ad*

*libitum* one hour prior to the experiment. Then, its pelvic waist was pressed to expel the urine and to obtain its 100% hydration level in relation to its standard body mass. We heated each individual inside a metal box wrapped in a thermal resistance for heating. The box had a movable lid, allowing the animal to easily leave the box when needed. A thin thermocouple (type-T, Omega®) was located in the inguinal region of each individual to record its body temperature during the heating [22]. Another type-T thermocouple was placed inside (on the surface) of the box to record heating rate of individuals. A dimmer previously connected to the box allowed to control that its temperature not exceeded 5–6˚C the temperature of the individual, allowing the thermoregulation of individuals, and avoided thermal shock and/or a premature exit of the box by the frog (i. e. before $VT_{Max}$ is reached; [22]). The thermocouples were calibrated and connected to a FieldLogger PicoLog TC-08 to record temperature data every 10 seconds. The $VT_{Max}$ of each individual was recorded as its last body temperature at the time of leaving the box. Once its final body mass was measured, it was taken to a container with water for recovery. Furthermore, to control for a potential effect of photoperiod on behavioral thermal tolerances, we tested if the $VT_{Max}$ differed between different times of the day by testing half of the individuals of each species in different periods: 10:00 to 17:00 (daytime) and 19:00 to 00:00 (nighttime).

**Statistical analyzes.** We used Mann-Whitney U tests to compare the $VT_{Max}$, and experimental variables between species. Experimental variables were: period (day or night), duration of experiment, initial body mass, initial body temperature, and heating rate. To test for the effect of possible confounding experimental variables on the $VT_{Max}$, we constructed generalized least squares models for each species. We used the corrected Akaike Information Criterion (AICc) to select the model that best represented the effects of factors and their interactions on the $VT_{Max}$ of each species. Differences of two units in AIC (ΔAICc) were not considered to be different [36]. We considered the model with weighted AIC (wAICc) values close or equal to 1 to represent the strongest model. All statistical analyzes and plotting were performed in R 3.5.0 [37], with the nlme [38], ggplot2 [39] and AICcmodavg [40] packages.

**Distribution and habitat.** We used vouchered occurrence data for *P. cuvieri* (N = 163) and *P. nattereri* (N = 164) in the Cerrado from a distribution database built for another study [41]. We calculated and mapped the difference between the $VT_{Max}$ and maximum environmental temperature ($ET_{Max}$; Bio 5; 30 seconds or ~1 km resolution from WorldClim Vr. 2.0; [42]), for each occurrence point of each species in Cerrado; the $VT_{Max}$ was that obtained at Estação Ecológica de Santa Bárbara. We used a Mann-Whitney U test to compare $VT_{Max}$—$ET_{Max}$ of species occurrence records. All maps and GIS procedures were made in QGIS 3.12 [43]. We tested for differences between species in habitat use by comparing abundances in open ('campo cerrado', 'campo sujo', and 'campo limpo') and non-open habitats (gallery forest, 'cerradão' and cerrado *stricto sensu*; [44]) for communities within Cerrado where both species occur in sympatry, available in the literature [45–50]. We used PAST [51] to test for differences between the proportion of each species in open and non-open habitats with chi-square and Fisher Exact tests, the latter when at least one cell was < 5.

## Results

### Voluntary Thermal Maximum ($VT_{Max}$) and experimental conditions

We found that $VT_{Max}$ was significantly lower for *P. cuvieri* than for *P. nattereri* (Table 1; U = 51, p = 0.0013). We also found significant differences in initial body mass (Table 1; U = 0, p < 0.0001) between species, with *P. nattereri* being heavier. We did not find significant differences in start body temperatures (Table 1; U = 112, p = 0.3359), period of day (Table 1;

**Table 1. Variation of the VT$_{Max}$ and predictor variables for *P. cuvieri* and *P. nattereri* from Estação Ecológica de Santa Bárbara, state of São Paulo, Brazil.**

| Variable | *Physalaemus cuvieri* | | *Physalaemus nattereri* | |
|---|---|---|---|---|
| | Mean ± SD | Range | Mean ± SD | Range |
| VT$_{Max}$ | 30.20 ± 1.69˚C | 27.48–33.13˚C | 32.74 ± 2.14˚C | 29.59–36.71˚C |
| Day | 29.62 ± 1.48˚C | 27.48–31.94˚C | 34.18 ± 1.62˚C | 32.09–36.71˚C |
| Night | 30.69 ± 1.76˚C | 28.14–33.13˚C | 31.74 ± 1.96˚C | 29.59–34.97˚C |
| DOE | 27.85 ± 18.17 min | 6–86 min | 26.72 ± 20.07 min | 6–81 min |
| ST | 25.79 ± 1.18˚C | 22.95–27.0˚C | 26.41 ± 2.30˚C | 22.73–30.58˚C |
| IBM | 2.15 ± 0.72 g | 1.19–3.82 g | 7.27 ± 7.52 g | 4.86–32.45 g |
| HRA | 0.07 ± 0.07˚C/min | 0.01–0.38˚C/min | 0.12 ± 0.21˚C/min | 0.06–0.84˚C/min |

Predictor variables are: period of day (day and night), initial body temperature (ST), duration of experiment (DOE), initial body mass (IBM), and heating rate (HRA).

U = 0.12, df = 32, p = 0.9051), duration of the experiment (Table 1; U = 128, p = 0.6872) and heating rate (Table 1; U = 123.5, p = 0.5752) between species (see S1, S2 and S3 Tables).

We compared six models for both species using the AIC selection criteria. For *P. nattereri*, the model including only period (day or night) was chosen as a better explanation of variation in the VT$_{Max}$ (Table 2), with higher values attained during daytime. For *P. cuvieri*, we retained the simpler null model, which showed a higher wAICc, which indicates that no variable explains the variation of the VT$_{Max}$ of this species (Table 3).

## Distribution and habitat

Overall distribution of occurrences was similar for the two species, occupying mainly the central and southern portions of the Cerrado (Fig 1; S4 Table). Thus, the distribution of

**Table 2. Effect of period, start body temperature, duration, initial body mass, and heating rate on the Voluntary Thermal Maximum (VT$_{Max}$) of *P. nattereri* from Estação Ecológica de Santa Bárbara, state of São Paulo, Brazil.**

| Model | Variables | Value | Std.Error | t-value | AICc | wAICc | ΔAICc |
|---|---|---|---|---|---|---|---|
| VI | Intercept | 34.245 | 0.7844 | 43.66 | 63.1 | 0.66 | 0.000 |
| | Period | -2.3937 | 1.0072 | -2.377 | | | |
| I | Intercept | 32.8771 | 0.5729 | 57.384 | 65.13 | 0.24 | 2.04 |
| V | Intercept | 33.48146 | 6.78518 | 4.934 | 67.13 | 0.09 | 4.03 |
| | Period | -2.35356 | 1.09774 | -2.144 | | | |
| | Start body temperature | 0.02784 | 0.24653 | 0.113 | | | |
| IV | Intercept | 33.402492 | 7.257777 | 4.602 | 72.18 | 0.01 | 9.08 |
| | Period | -2.375403 | 1.192946 | -1.991 | | | |
| | Start body temperature | 0.027744 | 0.258758 | 0.107 | | | |
| | Duration | 0.002234 | 0.029308 | 0.076 | | | |
| III | Intercept | 34.11138 | 7.23078 | 4.718 | 77.08 | 0 | 13.98 |
| | Period | -2.9531 | 1.29369 | -2.283 | | | |
| | Start body temperature | -0.03477 | 0.2628 | -0.132 | | | |
| | Duration | 0.01298 | 0.03072 | 0.422 | | | |
| | Initial body mass | 0.0873 | 0.08377 | 1.042 | | | |
| II | Intercept | 40.97635 | 8.8114 | 4.65 | 83.03 | 0 | 19.94 |
| | Period | -4.69461 | 1.85994 | -2.524 | | | |
| | Start body temperature | -0.28142 | 0.31754 | -0.886 | | | |
| | Duration | 0.04667 | 0.03908 | 1.194 | | | |
| | Initial body mass | 0.13547 | 0.08914 | 1.52 | | | |
| | Heating rate | -5.52697 | 4.19531 | -1.317 | | | |

**Table 3. Effect of period, start body temperature, duration, initial body mass, and heating rate on the Voluntary Thermal Maximum (VT$_{Max}$) of *P. cuvieri* from Estação Ecológica de Santa Bárbara, state of São Paulo, Brazil.**

| Model | Variables | Value | Std.Error | t-value | AICc | wAICc | ΔAICc |
|---|---|---|---|---|---|---|---|
| I | Intercept | 30.293 | 0.3788 | 79.98 | 81.52 | 0.48 | 0 |
| VI | Intercept | 29.69 | 0.5352 | 55.478 | 81.89 | 0.400 | 0.370 |
| | Period | 1.0964 | 0.7337 | 1.494 | | | |
| V | Intercept | 28.01163 | 8.54933 | 3.276 | 84.99 | 0.08 | 3.47 |
| | Period | 1.0601 | 0.7799 | 1.359 | | | |
| | Start body temperature | 0.06593 | 0.3355 | 0.196 | | | |
| IV | Intercept | 24.27443 | 8.89011 | 2.73 | 86.91 | 0.03 | 5.39 |
| | Period | 1.35975 | 0.81681 | 1.665 | | | |
| | Start body temperature | 0.15946 | 0.33822 | 0.471 | | | |
| | Duration | 0.02977 | 0.02327 | 1.279 | | | |
| III | Intercept | 24.16542 | 9.23459 | 2.617 | 91.07 | 0 | 9.55 |
| | Period | 1.4054 | 0.89665 | 1.567 | | | |
| | Start body temperature | 0.16829 | 0.35657 | 0.472 | | | |
| | Duration | 0.03163 | 0.02728 | 1.16 | | | |
| | Initial body mass | -0.09116 | 0.62658 | -0.145 | | | |
| II | Intercept | 24.89384 | 9.69648 | 2.567 | 95.73 | 0 | 14.21 |
| | Period | 1.38928 | 0.92409 | 1.503 | | | |
| | Start body temperature | 0.14817 | 0.37081 | 0.4 | | | |
| | Duration | 0.03157 | 0.02811 | 1.123 | | | |
| | Initial body mass | -0.05353 | 0.65185 | -0.082 | | | |
| | Heating rate | -2.10171 | 5.68863 | -0.369 | | | |

environmental temperatures was similar for both species. However, because the VT$_{Max}$ was different between species, the resulting distribution of VTMax—ET$_{Max}$ values was markedly different (Fig 1A and 1B). The north central portion of the Cerrado showed much higher environmental temperatures than the VT$_{Max}$ of *P. cuvieri* (Fig 1A), while this region is mostly below the VT$_{Max}$ of *P. nattereri* (Fig 1B). Furthermore, VTMax—ET$_{Max}$ values were found to be significantly different between species (U = 2249, p < 0.001; Fig 1C). *Physalaemus nattereri* is mostly found (~ 80%) on localities that attain maximum temperatures equal to or lower than its VT$_{Max}$, whereas *P. cuvieri* seems to be mostly distributed (~ 60%) in localities with temperatures higher than its VT$_{Max}$ (Fig 1C).

We obtained abundance data for five additional localities in southern Cerrado, most of them from protected areas (Fig 2; see also S2 Fig). In only two localities [49 and 50 + this study] we found significant differences between the proportion of each species in open and non-open habitats (S5 Table); in both cases, *P. cuvieri* was proportionally more abundant than *P. nattereri* in open areas. Considering the pooled abundances of these six studies, *P. cuvieri* was nearly twice more abundant in open (N = 2317 individuals) than in non-open areas (N = 1201), while *P. nattereri* was similarly abundant in open (N = 469) and non-open areas (N = 506; S5 Table). Furthermore, *P. cuvieri* was more abundant in open areas than in non-open areas in three localities and *P. nattereri*, in two localities, whereas both species were more abundant in non-open areas in two localities each (Fig 2; S5 Table).

## Discussion

Our results show that the Voluntary Thermal Maximum (VT$_{Max}$) is higher for *P. nattereri* than for *P. cuvieri*, contrary to our first prediction that larger body size (and an expected slower

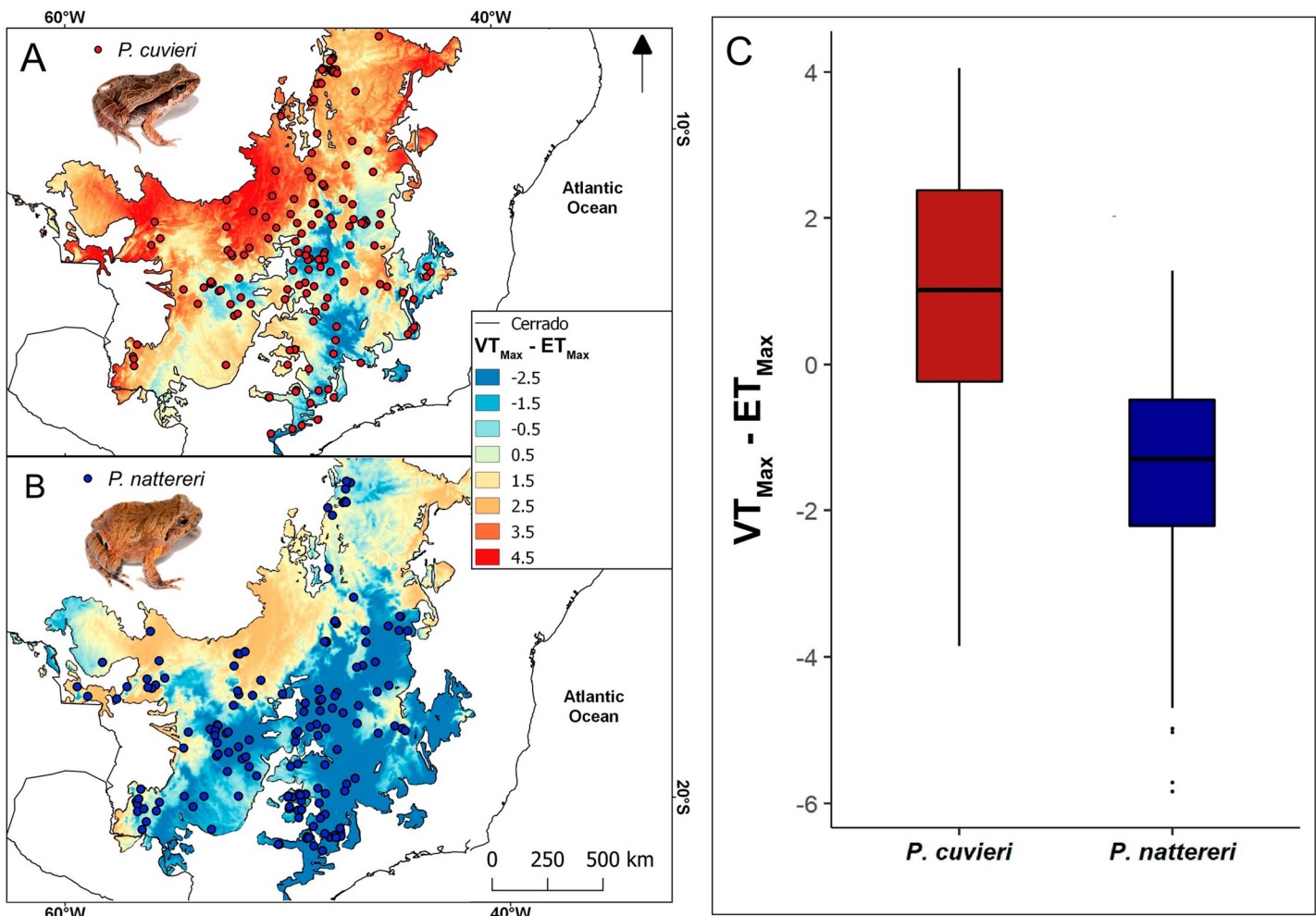

**Fig 1. Geographical distribution of the studied species and VT$_{Max}$—ET$_{Max}$ values throughout their distribution.** (A) Distribution of *Physalaemus cuvieri*; (B) distribution of *Physalaemus nattereri*; and (C) comparison of VT$_{Max}$—ET$_{Max}$ values at occurrence points between these species in the Cerrado.

cooling rate) would reflect in a lower VT$_{Max}$. Additionally, no difference in heating rate was found between species and only *P. nattereri* showed a significant difference on its VT$_{Max}$ between day and night. Regarding habitat use, in general, we found the species with lower VT$_{Max}$, *P. cuvieri*, to be more abundant in open habitats than in non-open habitats, which does not support our prediction that the species with the lower thermal tolerance should be less abundant in habitats with higher environmental temperatures. Lastly, in spite of both species being widespread in Cerrado, they showed different patterns of VT$_{Max}$—ET$_{Max}$ values throughout their ranges, with only *P. nattereri* having most of its records in localities with temperatures below its VT$_{Max}$. Thus, only for *P. nattereri* did we confirm our prediction that regional distribution comprises mostly localities with environmental temperatures below the VT$_{Max}$.

Regarding the lower VT$_{Max}$ values in the nocturnal period for *P. nattereri*, this result warrants future studies exploring variation in behavioral thermal tolerances in diurnal and nocturnal species in both periods of the day. Indeed, a higher VT$_{Max}$ during the day could reflect physiological adjustment of its thermal safety margin (see [22]), thus helping to protect the frog from extreme, potentially deleterious temperatures.

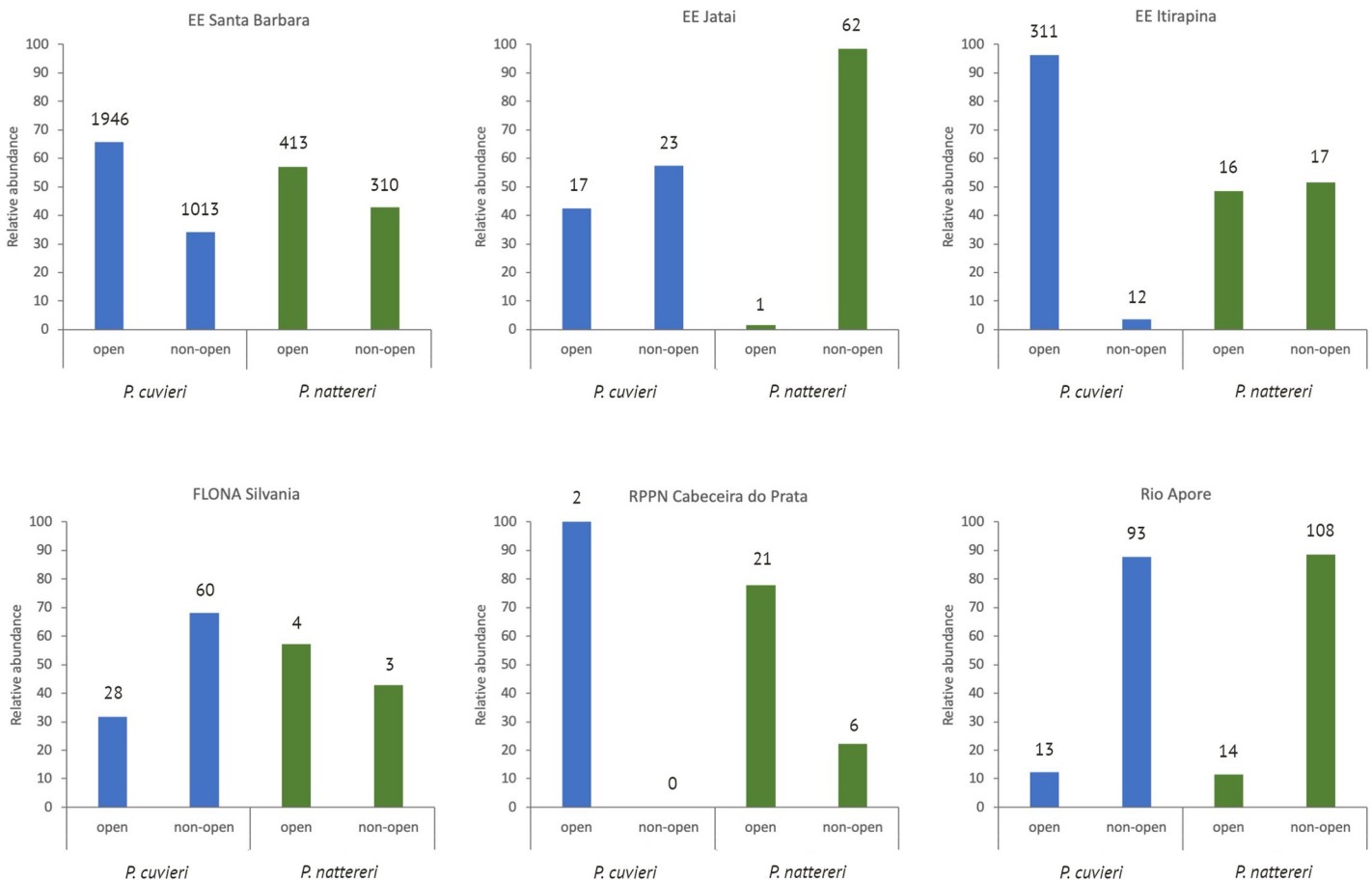

**Fig 2. Relative abundance (in %) of *P. cuvieri* (blue bars) and *P. nattereri* (green bars) in open and non-open areas in Cerrado (see S2 Fig).** Sources of data: [45–50]. Detailed data on the abundance of the frogs in different vegetation types are in S5 Table.

The difference in $VT_{Max}$ values between these two frog species might be related to their different body sizes [51, 52] but additionally might reflect their physiology and natural history. For instance, although there was no difference in heating rate between the species, *P. nattereri* might still cool slower when exposed to high temperatures because of its larger body size. As for differences in natural history, *P. nattereri* burrows in the soil [30, 31], which may allow it to quickly reduce its body temperature, since the soil is a good thermal insulant [53]. On the other hand, *P. cuvieri* uses pre-existing cavities as diurnal refuge (e. g. see [54]), which, in spite of also being below ground level (S3 Fig), are more exposed to variations in external environmental temperatures. Yet, despite having a lower $VT_{Max}$, most of the localities of *P. cuvieri* in Cerrado have temperatures above its $VT_{Max}$. This suggests that other aspects of its thermal ecology might be playing a role in avoiding thermal stress, such as a reduced daily activity time or physiological traits regulating hydration state.

As wet skin ectotherms, hydration level can also influence the temperatures tolerated and selected by individuals for thermoregulation in their habitats [55–58]. This has been observed for other frog species (e. g. *Rana catesbeiana*; [22]), with individuals decreasing their $VT_{Max}$ in response to dehydration, and some even losing their behavioral response to the $VT_{Max}$. Even though we controlled for hydration when measuring $VT_{Max}$, individuals in the wild rarely are at their optimal hydration level and thus desiccation might influence local frog distribution [59]. Desiccation has been shown to be correlated with substrate use [60] and with dispersal

probability throughout the landscape [59]. Additionally, closely related frog species may vary in their response to desiccation along thermal gradients, with some species showing greater resistance to water loss at lower temperatures, and others at higher temperatures [61]. Therefore, knowing the interaction between $VT_{Max}$ and hydration state of individuals in their environments can help to understand patterns and/or limits in their distribution [59, 62–64].

We found that *P. cuvieri*, the species with the lower $VT_{Max}$, was in general more abundant in open habitats, despite our second prediction that the species with the lower $VT_{Max}$ should be less abundant in warmer habitats (up to 35–37 °C in open habitats versus 32–35 °C in non-open habitats in our study area; pers. obs.). On the other hand, *P. nattereri*, which showed a higher $VT_{Max}$, was in general similarly abundant in open and non-open habitats. These results may reflect clade-related physiological constraints and further studies on the relationship of $VT_{Max}$ with habitat use should include additional species from both clades within the genus *Physalaemus* to which these species belong [28]. Although competition could also lead to differences in habitat use, especially in closely related species, we found no evidence of competition between our focal species in cerrado habitats (e. g. extensive niche overlap associated with limited resources, negative correlations between abundances; [65]).

Even though we found a relatively high variation in the data on habitat use for both species, the difference in the use of open and non-open habitats between species seems to be reflected in the overall patterns of their distribution throughout the Cerrado regarding their $VT_{Max}$. Indeed, *P. cuvieri* is in general more abundant in open and warmer habitats and occurs mostly in areas that attain maximum temperatures higher than its $VT_{Max}$, whereas *P. nattereri* tends to be abundant in both open and non-open (and cooler) areas and occurs mostly in areas that attain maximum temperatures below its $VT_{Max}$. Although geographic biases in sampling effort could affect these results, our study species are usually extremely abundant and conspicuous in localities where they occur, making them very easy to detect in inventories, by almost all frog sampling techniques. Thus, we are confident that the records in the maps of Fig 1 correspond to their overall actual distribution in the Cerrado. We highlight the importance of considering different spatial scales—geographic range and habitat use, as proposed by [66]—because these allow to quantify how species distribution may reflect different aspects of their niches.

Despite numerous ecophysiological studies comparing how environmental temperatures influence habitat use of species [11, 13], these rarely account for thermal tolerances. Using behavioral thermal tolerances, such as the $VT_{Max}$, allows for the integration of thermoregulatory behavior, which usually happens before critical limits are reached [3, 67, 68]. Furthermore, integrating the $VT_{Max}$ with natural history and geographic distribution data can be critical to understand how future scenarios of global warming might impact distribution [69, 70], especially for amphibians which are already under a global decline worldwide [71, 72]. Our study indicates that differences in behavioral thermal tolerance may be important in shaping local and regional distribution patterns. Furthermore, small-scale habitat use might reveal a link between behavioral thermal tolerance and natural history strategies. Further studies using additional sympatric species of the genus *Physalaemus* (e. g. *P. centralis*, from the same clade of *P. cuvieri*, and *P. marmoratus*, from the same clade of *P. nattereri*) could help to elucidate if those differences are due to body size variation or if tolerances are phylogenetically conserved. We hope this study stimulates future mechanistic studies on amphibian thermal ecology and on the impact of global warming on species distribution.

## Supporting information

**S1 Fig. Detail of hind feet of Physalaemus species in the study.** *P. nattereri* (A–B) and *P. cuvieri* (C–D), showing the inner and outer metatarsal tubercles in the detail. Note the much

larger and strongly keratinized tubercles in *P. nattereri*. Photos not to scale.
(PDF)

**S2 Fig. Relative abundance (in %) of *P. cuvieri* (blue circles) and *P. nattereri* (red circles) in open (brown) and non-open (green) areas in Cerrado (see S5 Table).** The localities are: Floresta Nacional (FLONA) de Silvânia (GO), Reserva Particular do Patrimônio Natural (RPPN) Cabeceira do Prata (MS), Estação Ecológica (EE) Jataí (SP), Estação Ecológica de Itirapina (SP), Estação Ecológica de Santa Bárbara (SP), and Aporé River (GO and MS). Sources of data: [45–50]. Detailed data on the abundance of the frogs in different vegetation types are in S5 Table.
(PDF)

**S3 Fig. Temperature during a 24-hour cycle measured in the field.** A) Temperature measured with sensors buried in the soil at superficial soil (green) and below ground level (red) and in a frog-sized plaster model (blue). B) Illustration of the measurement setup.
(PDF)

**S1 Table. Physiological data of species.** Data on each individual tested for Voluntary Thermal Maximum ($VT_{Max}$) in this study.
(XLSX)

**S2 Table. Temperature data of *P. cuvieri* during experiments.**
(XLSX)

**S3 Table. Temperature data of *P. nattereri* during experiments.**
(XLSX)

**S4 Table. Geographical records of both species in the Cerrado ecoregion.** Data from a distribution database built for another study [41].
(XLSX)

**S5 Table. Habitat and abundance data for both species in six localities of the Cerrado ecoregion.** [45–50].
(XLSX)

## Acknowledgments

We thank Instituto Florestal for allowing our work at Estação Ecológica de Santa Bárbara (permit #260108–008.476/2014; ICMBio-SISBIO for the permit to collect frog specimens (permit #50658–3). Paula Valdujo kindly provided occurrence data for both species. The comments and suggestions from an anonymous reviewer resulted in changes that enhanced the scientific quality of our manuscript.

## Author Contributions

**Conceptualization:** Juan C. Díaz-Ricaurte, Filipe C. Serrano, Marcio Martins.

**Data curation:** Juan C. Díaz-Ricaurte, Filipe C. Serrano, Marcio Martins.

**Formal analysis:** Juan C. Díaz-Ricaurte, Filipe C. Serrano, Estefany Caroline Guevara-Molina.

**Funding acquisition:** Marcio Martins.

**Investigation:** Juan C. Díaz-Ricaurte, Filipe C. Serrano, Marcio Martins.

**Methodology:** Juan C. Díaz-Ricaurte, Filipe C. Serrano.

**Project administration:** Juan C. Díaz-Ricaurte, Filipe C. Serrano, Marcio Martins.

**Resources:** Cybele Araujo, Marcio Martins.

**Supervision:** Marcio Martins.

**Validation:** Juan C. Díaz-Ricaurte, Filipe C. Serrano, Estefany Caroline Guevara-Molina, Marcio Martins.

**Visualization:** Juan C. Díaz-Ricaurte, Filipe C. Serrano, Estefany Caroline Guevara-Molina, Marcio Martins.

**Writing – original draft:** Juan C. Díaz-Ricaurte, Filipe C. Serrano.

**Writing – review & editing:** Juan C. Díaz-Ricaurte, Filipe C. Serrano, Estefany Caroline Guevara-Molina, Cybele Araujo, Marcio Martins.

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
