## [Decision Letter · Decision Letter 0]

5 Jun 2020

PONE-D-20-10531

Behavioral thermal tolerance predicts distribution pattern but not habitat use in sympatric Neotropical frogs

PLOS ONE

Dear Dr. Diaz-Ricaurte,

Thank you for submitting your manuscript to PLOS ONE. After careful consideration, we feel that it has merit but does not fully meet PLOS ONE’s publication criteria as it currently stands. Therefore, we invite you to submit a revised version of the manuscript that addresses the points raised during the review process.

There was one referee who found the paper to represent an interesting and relevant study. At the same time the referee found various issues which need to be solved including goal and hypothesis definition and analyses. I agree with this view and would like to encourage the authors to improve and re-submit their manuscript.  

We look forward to receiving your revised manuscript.

Kind regards,

Stefan Lötters

Academic Editor

PLOS ONE

Journal Requirements:

2. In your Methods section, please include a comment about the state of the animals following this research. Were they released, euthanized or housed for use in further research? If any animals were sacrificed by the authors, please include the method of euthanasia and describe any efforts that were undertaken to reduce animal suffering.

3. Your ethics statement must appear in the Methods section of your manuscript. If your ethics statement is written in any section besides the Methods, please move it to the Methods section and delete it from any other section. Please also ensure that your ethics statement is included in your manuscript, as the ethics section of your online submission will not be published alongside your manuscript.

4. We note that Figures 1 and 2 in your submission contain [map/satellite] images which may be copyrighted. All PLOS content is published under the Creative Commons Attribution License (CC BY 4.0), which means that the manuscript, images, and Supporting Information files will be freely available online, and any third party is permitted to access, download, copy, distribute, and use these materials in any way, even commercially, with proper attribution. For these reasons, we cannot publish previously copyrighted maps or satellite images created using proprietary data, such as Google software (Google Maps, Street View, and Earth). For more information, see our copyright guidelines: http://journals.plos.org/plosone/s/licenses-and-copyright.

1.    You may seek permission from the original copyright holder of Figures 1 and 2 to publish the content specifically under the CC BY 4.0 license.

Additional Editor Comments (if provided):

Reviewers' comments:

Reviewer's Responses to Questions

**Comments to the Author**

1. Is the manuscript technically sound, and do the data support the conclusions?

Reviewer #1: Yes

2. Has the statistical analysis been performed appropriately and rigorously? 

Reviewer #1: Yes

3. Have the authors made all data underlying the findings in their manuscript fully available?

Reviewer #1: Yes

4. Is the manuscript presented in an intelligible fashion and written in standard English?

Reviewer #1: Yes

5. Review Comments to the Author

Reviewer #1: The manuscript investigates whether behavioral tolerances (VTmax) of two anurans from the Brazilian savanna differ. Additionally, the authors explore the potential consequences of VTmax on the geographical distribution patterns and habitat use of these two species. These topics are relevant across the ecology of anuran species from the Brazilian savanna since it is a highly threatened biome, and information on physiology, behavior, and distribution about its resident species are currently scarce. Overall, I enjoyed reading the manuscript, and I believe the topic itself is absolutely enthralling.

However, the relevant aspects of the manuscript need further clarification and details. Below, I list four significant issues I found in the paper that hindered its full understanding.

1) In the introduction section, I have the impression that the authors accidentally confound their predictions with the hypothesis itself. I guess there must be a hypothesis behind the authors’ main question. I also noted that the hypothesis is not clearly stated in the paragraph where authors present their goals to the reader (Body size affects/is correlated to behavioral thermal tolerances). I also find the authors’ predictions about VTmax hard to follow. A schematic figure of how predictors influence VTmax may be needed.

2) The introduction does not properly focus on what behavioral thermal tolerance is and its link to species distribution and habitat use. I suggest the authors primarily focus on introducing physiological aspects of thermal tolerance and then explore how they potentially affect species distribution patterns, abundances, and habitat use rather than focusing on the biome and taxonomic aspects. By doing so, the introduction can be more thorough and would give a better overview of the topic to the readers.

3) Statistical analysis needs a more detailed description. The authors do not justify why they opted for building and competing gls models. I wonder whether the data was heteroskedastic. I am also concerned about the fact that room temperature was not controlled even though the authors tried to control for a potential circadian effect. However, I am not sure whether grouping experiments performed at 10 a.m. with those performed at 5 p.m. (if I understood the author’s idea of controlling for circadian effects) is the best approach. I am not also sure whether controlling for “circadian effects” is the correct term. I guess the authors aimed to incorporate potential differences in room temperature along the day of the experiment and referred to such effects as circadian effects. Moreover, I wonder whether it is necessary to perform posthoc tests to confirm the VTmax differences found between species.

53 Apparently, references 4, 5, 6 goes better with the statement in lines 50-51

55 I believe this reference is not appropriate.

58 how about local factors such as increased edge effects due to landscape modification (i.e. habitat fragmentation). I also recommend you to check the work of Stillman 2019.

59 points to the limitation of modelling species based only on abiotic factors but present solution right aaway

62 what is behavioral tolerance?

64 whtat is thermal tolerance treshold?

68 define PBT

69 long sentence

77 Reference number 15 is not published. Why did you use it?

80 This sentence should be placed with the one in line 59.

83 I think the reason why Vtmax is an informative was explained preivously but it is not properly linked to them. Consider revising this paragraph

86 include reference about the Vtmax not affecting locomotion

103 previously-dug (excavated) burrows

106 the paragraph is not clear enough. Not sure if the authors indeed tested for effect of temprature (their goal is much more interesting than simply testing the effects of temperature on distribution and behavioral thermal tolerances

107 adopt only one term (global or regional distribution pattern)

110 They are not hypotehsis but predictions

111 cooling rates? I thought you measured heating rates. (I think heating rates would be the appropiate term and it should be standardized over the manuscript). Furthermore, I could not follow your predictions. I think you need to clarify them.

112 since you expect P. nattereri to have a lower Vtmax than P. cuvieri, why you did not mention P. nattereri in the second prediction?

115 how? too simplistic sentence compared to the considerable importance of the study

137 unclear sentence. I did not understand. Please, provide more details.

146 specify why exceeding 5-6°C would be a problem

152 12h format instead of 24h format

156 was the data heteroskedastic?

173 The acronym ET meaning was not introduced before.

174 Please provide the equivalent resolution in Km

177 It is not clear from where you obtained abundance data.

187 Please specify which species is larger

195 Experimental variables or predictor variables?

197 IBM Range seems to be in the wrong format.

199 I am afraid you did not test models. We generally build and compete models.

207 Table 2 - change table, there are two different statistical approaches. You either use one or another. See the comments below.

211 Table 3 - change table, there are two different statistical approaches. You either use one or another. See the comments below.

237 Which statistical test was used to test differences in abundance among habitats? I could not find it in the material & methods section

246 The area is currently a National Forest (FLONA)

254 cooling rates or heating rates? You should standardize the term. You did not measure cooling rates.

256 How would this affect your results? I am not sure if you approach this topic in the discussion section.

263 Concordance - review English. Additionally, I think you should also use the term regional rather than global. The species studied are restricted to the neotropics.

291 this was not formally tested

General comments and questions

Discussion

How may the sex of individuals have influenced the results? Have you performed the experiments only with males, or were there also females? I think the authors should include this information

Table 1 & 2

Remove p-values from the table that show model parameters. P-values are pertinent in the frequentist approach and should not be used along with information theory approaches. You must also include ΔAIC and organize table from the lowest to the highest values.

Supplementary material

Please, provide a description of each term in the columns.

How would competition influence may influence the results found? This should be discussed in the discussion section if these species compete (which I think it is possible since competition may be stronger between congeners.)

6. PLOS authors have the option to publish the peer review history of their article (what does this mean?). If published, this will include your full peer review and any attached files.

Reviewer #1: No

---

## [Author Response · Author response to Decision Letter 0]

16 Jul 2020

Dear Dr. Stefan Lötters

Academic Editor

PLOS ONE

We thank the Academic Editor and reviewer #1 for their time and attention in reviewing our manuscript. The comments and suggestions provided reviewer #1 resulted in changes that enhanced the scientific quality of our manuscript. We have accepted all of the grammatical changes and suggestions. For each comment, we made all the respective corrections, with detailed answers specifying the number of the lines in which the text was changed. If you find that our response is not clear, you can also check the changes in the manuscript version with track changes.

COMMENTS FROM EDITOR:

Comment 1: Please ensure that your manuscript meets PLOS ONE's style requirements, including those for file naming. The PLOS ONE style templates can be found at https://journals.plos.org/plosone/s/file?id=wjVg/PLOSOne_formatting_sample_main_body.pdf and https://journals.plos.org/plosone/s/file?id=ba62/PLOSOne_formatting_sample_title_authors_affiliations.pdf

R: We followed all style requirements, including those for file name.

Comment 2: In your Methods section, please include a comment about the state of the animals following this research. Were they released, euthanized or housed for use in further research? If any animals were sacrificed by the authors, please include the method of euthanasia and describe any efforts that were undertaken to reduce animal suffering. 

R: Done. We added this information at the end of the second paragraph of the Methods section. See lines 147-149: “…This study was conducted under a permit by Comissão de Ética no Uso de Animais (CEUA #2325141019) of Instituto Butantan. All animals were alive after the experiments described below and were released the following morning at the site of capture.”

Comment 3: Your ethics statement must appear in the Methods section of your manuscript. If your ethics statement is written in any section besides the Methods, please move it to the Methods section and delete it from any other section. Please also ensure that your ethics statement is included in your manuscript, as the ethics section of your online submission will not be published alongside your manuscript.

R: We added the ethics statement at the end of the second paragraph of the Methods section. See the above comment.

Comment 4: We note that Figures 1 and 2 in your submission contain [map/satellite] images which may be copyrighted. All PLOS content is published under the Creative Commons Attribution License (CC BY 4.0), which means that the manuscript, images, and Supporting Information files will be freely available online, and any third party is permitted to access, download, copy, distribute, and use these materials in any way, even commercially, with proper attribution. For these reasons, we cannot publish previously copyrighted maps or satellite images created using proprietary data, such as Google software (Google Maps, Street View, and Earth). For more information, see our copyright guidelines: http://journals.plos.org/plosone/s/licenses-and-copyright.

R: Our figures were built by us in QGIS and no satellite image or other copyrighted material was used to build them.

REVIEWERS' COMMENTS:

Reviewer 1:

General considerations: 

The manuscript investigates whether behavioral tolerances (VTmax) of two anurans from the Brazilian savanna differ. Additionally, the authors explore the potential consequences of VTmax on the geographical distribution patterns and habitat use of these two species. These topics are relevant across the ecology of anuran species from the Brazilian savanna since it is a highly threatened biome, and information on physiology, behavior, and distribution about its resident species are currently scarce. Overall, I enjoyed reading the manuscript, and I believe the topic itself is absolutely enthralling. However, the relevant aspects of the manuscript need further clarification and details. Below, I list four significant issues I found in the paper that hindered its full understanding.

Comment 1: In the introduction section, I have the impression that the authors accidentally confound their predictions with the hypothesis itself. I guess there must be a hypothesis behind the authors’ main question. I also noted that the hypothesis is not clearly stated in the paragraph where authors present their goals to the reader (Body size affects/is correlated to behavioral thermal tolerances). I also find the authors’ predictions about VTmax hard to follow. A schematic figure of how predictors influence VTmax may be needed.

R: We thank the reviewer for calling our attention for this problem. We reorganized the introduction and rephrased some portions (especially the last paragraph) in order to make clear our question, hypothesis, and predictions. We also made clear, in the Methods section, that we tested the possible effects of confounding experimental variables on VTMax.

Comment 2: The introduction does not properly focus on what behavioral thermal tolerance is and its link to species distribution and habitat use. I suggest the authors primarily focus on introducing physiological aspects of thermal tolerance and then explore how they potentially affect species distribution patterns, abundances, and habitat use rather than focusing on the biome and taxonomic aspects. By doing so, the introduction can be more thorough and would give a better overview of the topic to the readers.

R: We thank the reviewer for calling our attention for this problem. We changed the introduction section to make it clearer and more direct. 

Comment 3: Statistical analysis needs a more detailed description. The authors do not justify why they opted for building and competing gls models. I wonder whether the data was heteroskedastic. I am also concerned about the fact that room temperature was not controlled even though the authors tried to control for a potential circadian effect. However, I am not sure whether grouping experiments performed at 10 a.m. with those performed at 5 p.m. (if I understood the author’s idea of controlling for circadian effects) is the best approach. I am not also sure whether controlling for “circadian effects” is the correct term. I guess the authors aimed to incorporate potential differences in room temperature along the day of the experiment and referred to such effects as circadian effects. Moreover, I wonder whether it is necessary to perform posthoc tests to confirm the VTmax differences found between species.

R: Changed as suggested. We tried to make the text on statistical methods clearer. We used GLS models to test for the effect of possible confounding experimental variables on the VTMax. We tested all data for heteroscedasticity, before choosing the tests. Furthermore, we changed the t-test for the non-parametric Mann-Whitney U-tests when necessary. As for the time when experiments were carried out, we grouped data in two categories only (day and night) because these periods are different in terms of air humidity, temperature etc., and we wanted to test whether tests made at different periods would result in differences in VTMax. We also removed the term circadian from the text. Finally, since we compared only two species, there was no reason to perform post-hoc tests. 

Specific comments:

Comment 4-Line 53: Apparently, references 4, 5, 6 goes better with the statement in lines 50-51.

R: Changed as suggested. We additionally added a more adequate reference to the following sentence; see lines 59-60: “…Several studies have explored environmental constraints on ectothermic vertebrates at regional and global scale [1, 4]…”

Comment 5-Line 55: I believe this reference is not appropriate.

R: We agree with the reviewer. We replaced it with two more adequate references, see line 62: “…which can lead to declining populations and/or local extinctions [6–7]…”

Comment 6-Line 58: How about local factors such as increased edge effects due to landscape modification (i.e. habitat fragmentation). I also recommend you to check the work of Stillman 2019. 

R: We thank the reviewer for the suggestion. We added both habitat fragmentation and heat waves to the climate change scenario. See lines 65-66: “…especially under a current global warming (e.g. [8–9), as well as habitat disturbance causing microclimate changes (e.g. habitat fragmentation; [10])…”

Comment 7-Line 59: Points to the limitation of modelling species based only on abiotic factors but present solution right away.

R: Here we aimed to show that incorporating thermal tolerances to climatic niche modeling allows a better estimate of potential distributions. We made important changes in this paragraph. Please, see from lines 67 to 77. 

Comment 8-Line 62: What is behavioral tolerance?

R: We here refer to behavioral thermal tolerance. To make the text clearer, we added a sentence defining behavioral and physiological tolerances; see from lines 82-91: “…Identifying thermal tolerance thresholds (i.e. measurable thermal limits) outside the range of preferred body temperatures (PBT) for thermoregulation (see [18]) allows for the identification of temperatures that directly affect the behavioral and physiological thermal tolerance of ectothermic organisms. One of the thresholds related to PBT is the Voluntary Thermal Maximum (VTMax), which represents a behavioral thermal tolerance. VTMax is the maximum temperature that an organism will endure before trying to move to a place with a lower temperature, thus trying to maintain its body temperature within its range of PBT [3, 18–19]. If an individual fails to respond to its VTMax, an increase in body temperature will expose it to its physiological thermal limit (i.e. its Critical Thermal Maximum), which can lead to functional collapse and consequently death due to overheating [19–20]…”

Comment 9-Line 64: What is thermal tolerance threshold?

R: We added a brief definition: measurable thermal limits. Please, see the above comment.

Comment 10-Line 68: Define PBT.

R: Done. We changed PBT to Preferred body temperatures. See lines 82-83: “…Identifying thermal tolerance thresholds (i.e. measurable thermal limits) outside the range of preferred body temperatures (PBT) for thermoregulation (see [18])…”

Comment 11-Line 69: Long sentence.

R: We divided the sentence into two distinct sentences; see lines 86-91: “…VTMax is the maximum temperature that an organism will endure before trying to move to a place with a lower temperature, thus trying to maintain its body temperature within its range of PBT [3, 18–19]. If an individual fails to respond to its VTMax, an increase in body temperature will expose it to its physiological thermal limit (i.e. its Critical Thermal Maximum), which can lead to functional collapse and consequently death due to overheating [19–20]…”

Comment 12-Line 77: Reference number 15 is not published. Why did you use it?

R: This publication, which is the only article which tests the impact of hydration status on VTmax, is currently in the second review cycle at the Journal of Thermal Biology. Only minor revisions were asked in this cycle. Thus, it may be at least online first when the proof reading of the current manuscript would be made (provided it is accepted). However, we still can cite the Master’s thesis that originated the article in our final manuscript (this is what we made in the current version of our manuscript).

Comment 13-Line 80: This sentence should be placed with the one in line 59.

R: While both sentences call attention to the potentially detrimental effects of warming and heating to ectotherms, line 59 (now line 67) does so at the species or population level, while line 80 (now line 101) does so at the individual level. We hope this statement clarifies the reason to maintain these sentences where they are.

Comment 14-Line 83: I think the reason why Vtmax is an informative was explained previously but it is not properly linked to them. Consider revising this paragraph

R: We agree. We moved the text in lines 83-87 to the previous paragraph. See lines 91-96: “…Therefore, the behavioral response to upper limits might represent a more informative ecological threshold to identify thermal constraints on habitat use and geographic distribution [3, 8, 13]. Contrary to the Critical Thermal Maximum, the exposure to the VTMax does not induce an immediate loss of locomotion [3, 21]. Therefore, VTMax can more realistically portrait changes in species behavior associated with their natural history…”

Comment 15-Line 86: Include reference about the Vtmax not affecting locomotion

R: We agree with the reviewer’s comment and added two references that show how VTmax does not affect locomotion. See lines 94-95: “…the exposure to the VTMax does not induce an immediate loss of locomotion [3, 21]…”

Comment 16-Line 103: Previously-dug (excavated) burrows.

R: Change as suggested. See line 130: “…during the day in previously-dug burrows, P. nattereri breeds mostly in temporary puddles…”

Comment 17-Line 106: The paragraph is not clear enough. Not sure if the authors indeed tested for effect of temperature (their goal is much more interesting than simply testing the effects of temperature on distribution and behavioral thermal tolerances.

R: We thank the reviewer for the comments and have rephrased the paragraph to better reflect the importance and predictions of our study; see from lines 103 to 115.

Comment 18-Line 107: Adopt only one term (global or regional distribution pattern).

R: Done. See lines 103-104: “…regional distribution patterns…”

Comment 19-Line 110: They are not hypothesis but predictions

R: We agree with the reviewer’s comments and have changed the text accordingly. See lines 105-112: “…Our hypothesis is that, by being a measure that reflects avoidance of stressful thermal conditions, VTMax determines both habitat use and geographic distribution in these species. If VTMax decreases with body size (see above; [23–24]), we predict that VTMax is lower in the larger species. Furthermore, if VTMax determines habitat use and geographic distribution, we predict that (i) the species with lower VTMax is less abundant in habitats with higher environmental temperatures and that (ii), regarding geographic distribution, both species occur mostly in localities where the maximum environmental temperature is below their VTMax…”

Comment 20-Line 111: Cooling rates? I thought you measured heating rates. (I think heating rates would be the appropriate term and it should be standardized over the manuscript). Furthermore, I could not follow your predictions. I think you need to clarify them.

R: We agree with the reviewer that “cooling rates” could confuse the reader. Although we believe that post-heating stress might play a role in thermal tolerance, we did not test them and thus we rephrased our predictions to make them clearer. Please, see the above comment.

Comment 21-Line 112: Since you expect P. nattereri to have a lower Vtmax than P. cuvieri, why you did not mention P. nattereri in the second prediction?

R: We agree with the reviewer’s comments and have changed the text accordingly. Please, see the above comment.

Comment 22-Line 115: How? too simplistic sentence compared to the considerable importance of the study.

R: We agree with the reviewer’s comments, and have changed the text to better reflect the importance of our study. See lines 112-115: “…We expect that our results can contribute to assess the vulnerability of Neotropical frogs to climate change by integrating their behavioral thermal tolerances with their habitat use and distribution patterns, in order to identify areas with potential stressful climatic conditions to their populations…”

Comment 23-Line 137: Unclear sentence. I did not understand. Please, provide more details.

R: We have rephrased it and made the sentence clearer. See lines 152-154: “…To obtain the VTMax for each species, we measured each individual at 100% hydration level less than 24 hours after capture. To reach maximum hydration level, each individual was placed in a cup with water ad libitum one hour prior to the experiment…”

Comment 24-Line 146: Specify why exceeding 5-6°C would be a problem.

R: The VTMax measuring method attempts to maintain a gradual and constant heating rate of the individual in relation to the heating rate of the box. Exposing the individual to high temperatures (over 5 or 6ºC higher than its body temperature) can lead to thermal shock. The gradual heating between the individual and the box allows for the animal to thermoregulate until it makes the decision to leave the box. In other studies (e.g. Recoder et al 2018; Guevara-Molina 2019; Diaz-Ricaurte and Serrano 2020) maintaining this temperature range has prevented the individual from prematurely exiting the box.

Comment 25-Line 152: 12h format instead of 24h format.

R: We changed these times to the format we found in two PlosOne papers (https://doi.org/10.1371/journal.pone.0219907 and ttps://doi.org/10.1371/journal.pone.0077902).

Comment 26-Line 156: Was the data heteroskedastic?

R: The data is indeed heteroskedastic. One of the reasons we used GLS is because they can handle heteroskedastic data better than other methods. We thank the reviewer for further bringing this into our attention. To improve the validity of our results, we weighed the squared residuals of the regression in each model to better account for data variance. Although the results did not change, the heteroskedasticity markedly decreased and our results are now statistically sound.

Comment 27-Line 173: The acronym ET meaning was not introduced before.

R: We changed the text to include the meaning of ET. See lines 187-188: “…the difference between the VTMax and maximum environmental temperature (ETMax; Bio 5; 30 seconds or ~1 km resolution from WorldClim Vr. 2.0; [42])…”

Comment 28-Line 174: Please provide the equivalent resolution in Km.

R: We added the km resolution in the text. See line 188: “…30 seconds or ~1 km resolution from WorldClim…”

Comment 29-Line 177: It is not clear from where you obtained abundance data.

R: We clarified in the text that we obtained abundance data from the literature and added the sources. See lines 191-195: “…We tested for differences between species in habitat use by comparing abundances in open (‘campo cerrado’, ‘campo sujo’, and ‘campo limpo’) and non-open habitats (gallery forest, ‘cerradão’ and cerrado stricto sensu; [44]) for communities within Cerrado where both species occur in sympatry, available in the literature [45–50]…”

Comment 30-Line 187: Please specify which species is larger.

R: We have added in the text that P. nattereri was heavier than P. cuvieri. See lines 122-126: “…Physalaemus nattereri has a stout body, a moderate to large size (adult snout-to-vent length of 29.8–50.6 mm) and is endemic to the Cerrado, whereas P. cuvieri has a slenderer body, a smaller size (snout-to-vent length of adults 28–30 mm) and occurs throughout the Cerrado, in southern portions of the Amazon Forest and in the Atlantic Forest [29])…”

Comment 31-Line 195: Experimental variables or predictor variables?

R: Changed as suggested. See the legend of Table 1 in lines 208-211: “…Table 1. Variation of the VTMax and predictor variables for P. cuvieri and P. nattareri from Estação Ecológica de Santa Bárbara, state of São Paulo, Brazil. Predictor variables are: period of day (day and night), initial body temperature (ST), duration of experiment (DOE), initial body mass (IBM), and heating rate (HRA).”

Comment 32-Line 197: IBM Range seems to be in the wrong format.

R: Thank you, we corrected it in Table 2. We additionally clarified that two of the columns refer to “Mean ± SD”.

Comment 33-Line 199: I am afraid you did not test models. We generally build and compete models.

R: We agree with the reviewer and have replaced “tested” with “compared”. See line 214: “We compared six models for both species using the AIC selection criteria…”

Comment 34-Line 207: Table 2 - change table, there are two different statistical approaches. You either use one or another. See the comments below.

R: We agree with the reviewer’s comments and thus removed the column containing p-values.

Comment 35-Line 211: Table 3 - change table, there are two different statistical approaches. You either use one or another. See the comments below.

R: Done. We removed the column containing p-values.

Comment 36-Line 237: Which statistical test was used to test differences in abundance among habitats? I could not find it in the material & methods section.

R: We did not perform any statistical test. Now we used chi-square and Fisher exact tests to test for differences between the proportion of each species in open and non-open habitats. The Material & Methods section, the paragraph on these data in the Results section, and S5 Table were modified accordingly.

Comment 37-Line 246: The area is currently a National Forest (FLONA).

R: Thank you for bringing this into our attention. We changed the legend of figure 2. See lines 262-268: “Fig 2. Relative abundance (in %) of P. cuvieri (blue circles) and P. nattereri (red circles) in open (brown) and non-open (green) areas in Cerrado (see S5 Table). The localities are: Floresta Nacional (FLONA) de Silvânia (GO), Reserva Particular do Patrimônio Natural (RPPN) Cabeceira do Prata (MS), Estação Ecológica (EE) Jataí (SP), Estação Ecológica de Itirapina (SP), Estação Ecológica de Santa Bárbara (SP), and Aporé river (GO and MS). Sources of data: [45–50]. Detailed data on the abundance of the frogs in different vegetation types are in S5 Table…”

Comment 38-Line 254: Cooling rates or heating rates? You should standardize the term. You did not measure cooling rates.

R: Even though we only estimated heating rates, cooling rates are part of the argument on why thermal inertia may make larger animals to have lower VTMax. We argue that after reaching VTMax, larger animals such as P. nattereri take longer to cool down and this may expose it to a risk of overheating for a longer period. We include this argument to justify why larger animals have lower VTMax, despite only measuring heating rates.

Comment 39-Line 256: How would this affect your results? I am not sure if you approach this topic in the discussion section. 

R: If this comment is related to the previous sentence, regarding the difference in VTMax between day and night we found for P. nattereri, we think it is an interesting avenue for future studies since night temperatures are frequently not considered in studies of thermal tolerances. We discuss this subject in lines 274-275: “…only P. nattereri showed a significant difference on its VTMax between day and night…” and in lines 284-288: “…Regarding the lower VTMax values in the nocturnal period for P. nattereri, this result warrants future studies exploring variation in behavioral thermal tolerances in diurnal and nocturnal species in both periods of the day. Indeed, a higher VTMax during the day could reflect physiological adjustment of its thermal safety margin (see [22]), thus helping to protect the frog from extreme, potentially deleterious temperatures.” 

We thank the reviewer for calling our attention to this aspect of our study.

Comment 40-Line 263: Concordance - review English. Additionally, I think you should also use the term regional rather than global. The species studied are restricted to the neotropics.

R: We replaced “global” with “regional” and corrected the concordance throughout the text.

Comment 41-Line 291: This was not formally tested.

R: The new version of the manuscript includes tests regarding abundances in open and non-open habitats. See lines 195-197: “…We used PAST [51] to test for differences between the proportion of each species in open and non-open habitats with chi-square and Fisher Exact tests, the latter when at least one cell was < 5…”

Comment 42: How may the sex of individuals have influenced the results? Have you performed the experiments only with males, or were there also females? I think the authors should include this information.

R: Unfortunately, we were unable to determine sex for most individuals and thus did not include this variable in our models. Because these species usually show sexual dimorphism, with females being larger than males, we believe that using body mass as a variable in our models accounts for this. Furthermore, for both species, the models including “Initial body mass” were consistently the least supported ones, which might indicate that this potential difference between sexes is not a significant factor to affect the VTMax.

Comment 43-Tables 1 & 2: Remove p-values from the table that show model parameters. P-values are pertinent in the frequentist approach and should not be used along with information theory approaches. You must also include ΔAIC and organize table from the lowest to the highest values.

R: We agree with the reviewer and have removed the p-values from the tables. We also included ΔAIC and organized both tables from the lowest to the highest values.

Comment 44-Supplementary material: Please, provide a description of each term in the columns.

R: Done. We provided the details of each column in supplementary tables.

Comment 45: How would competition influence may influence the results found? This should be discussed in the discussion section if these species compete (which I think it is possible since competition may be stronger between congeners.).

R: An evidence of competition between these species would be a checkerboard distribution in different habitats or localities and this is not evident neither in our data nor in our personal experience with these animals. Furthermore, there is no evidence that any resource is limited for this species in their cerrado habitats or in their breeding habitats (they don't even overlap much in their breeding habitats or strategy). Even so, we included a short discussion on this possibility in the Discussion; see from lines 317 to 328.

Finally, we added a few lines to the discussion and to the abstract highlighting our finding that the difference in the use of open and non-open habitats between species seems to reflect the overall patterns of their distribution throughout the Cerrado regarding their VTmax; see from lines 328 to 341.

---

## [Decision Letter · Decision Letter 1]

25 Aug 2020

PONE-D-20-10531R1

Does behavioral thermal tolerance predict distribution pattern and habitat use in two sympatric Neotropical frogs?

PLOS ONE

Dear Dr. Diaz-Ricaurte,

Thank you for submitting your manuscript to PLOS ONE. After careful consideration, we feel that it has merit but does not fully meet PLOS ONE’s publication criteria as it currently stands. Therefore, we invite you to submit a revised version of the manuscript that addresses the points raised during the review process.

The current version is much better than the previous one and is almost acceptable. There were few minor comments for your consideration made by the referee.

We look forward to receiving your revised manuscript.

Kind regards,

Stefan Lötters

Academic Editor

PLOS ONE

Reviewers' comments:

Reviewer's Responses to Questions

**Comments to the Author**

1. If the authors have adequately addressed your comments raised in a previous round of review and you feel that this manuscript is now acceptable for publication, you may indicate that here to bypass the “Comments to the Author” section, enter your conflict of interest statement in the “Confidential to Editor” section, and submit your "Accept" recommendation.

Reviewer #1: All comments have been addressed

2. Is the manuscript technically sound, and do the data support the conclusions?

Reviewer #1: Yes

3. Has the statistical analysis been performed appropriately and rigorously? 

Reviewer #1: Yes

4. Have the authors made all data underlying the findings in their manuscript fully available?

Reviewer #1: Yes

5. Is the manuscript presented in an intelligible fashion and written in standard English?

Reviewer #1: Yes

6. Review Comments to the Author

Reviewer #1: The authors have satisfactorily addressed the comments from the previous review round. The manuscript is thorough and clear with the exception of very specific parts. Below are my comments on the aspects that must be improved.

Discussion section

Maybe the authors could also discuss how sampling bias can affect the results. This is because there is a sampling bias towards the southern portion of the Cerrado biome (see Diniz-Filho et al. 2005; Costa et al. 2007)

Diniz-Filho et al. 2005 DOI: 10.1111/j.1466-822x.2005.00165.x

Costa et al. 2007 DOI: 10.1111/j.1472-4642.2007.00369.x

Line 60: regional and global scales

Line 65: the current global warming crisis

Line 86: a behavioral thermal tolerance measure

Line 95: portray

Line 105: for being...

Lines 107-112: Why not to mention the species that thte authors know has a larger body size (P. nattereri). I also wonder why not directly referring to "open habitats" as having higher environmental temperature.

Line 128: These two species

Line 142: The landscape not only consist of...

Line 191: occurence records

Lines 217: The model with the highest weight was the null model. I suggest the authors to rephrase this sentence so that it sound more clear and honest.

Line 235: was markedly

Line 241-243: It is not clear whether P. cuvieri occur in areas where the maximum temperature is lower or higher thans its VTmax

Lines 250-259: The results presentend in this part of the text should be summarized through charts. It was difficult to withdraw conclusions from Figure 2. I suggest that Figure 2 should be included in the Supplementary material and a chart (barplots or boxplots) should be presented along the manuscript instead.

Line 330: tend to be abundant in both open and non-open areas

7. PLOS authors have the option to publish the peer review history of their article (what does this mean?). If published, this will include your full peer review and any attached files.

Reviewer #1: No

---

## [Author Response · Author response to Decision Letter 1]

29 Aug 2020

Dear Dr. Stefan Lötters

Academic Editor

PLOS ONE

We thank the Academic Editor and reviewer #1 for their time and attention in reviewing our manuscript. The comments and suggestions provided by reviewer #1 resulted in changes that enhanced the scientific quality of our manuscript. We have accepted all of the grammatical changes and suggestions. For each comment, we made all the respective corrections, with detailed answers specifying the number of the lines in which the text was changed. If you find that our response is not clear, you can also check the changes in the manuscript version with track changes.

REVIEWERS' COMMENTS:

Reviewer 1:

General considerations: 

The authors have satisfactorily addressed the comments from the previous review round. The manuscript is thorough and clear with the exception of very specific parts. Below are my comments on the aspects that must be improved.

Comment 1: Maybe the authors could also discuss how sampling bias can affect the results. This is because there is a sampling bias towards the southern portion of the Cerrado biome (see Diniz-Filho et al. 2005; Costa et al. 2007).

R: We thank the reviewer for calling our attention for this problem. We added a phrase in the discussion section to clarify that the distribution of our study species is unlikely to be prone to a strong sampling bias due to their easy detectability. See lines 337-341.

Specific comments:

Comment 2-Line 60: regional and global scales

R: Changed as suggested, see line 60.

Comment 3-Line 65: the current global warming crisis

R: Changed as suggested, see line 65.

Comment 4-Line 86: a behavioral thermal tolerance measure

R: Changed as suggested, see line 86.

Comment 5-Line 95: portray

R: Changed as suggested. see line 95.

Comment 6-Line 105: for being...

R: Changed as suggested, see line 105.

Comment 7-Lines 107-112: Why not to mention the species that thte authors know has a larger body size (P. nattereri). I also wonder why not directly referring to "open habitats" as having higher environmental temperature.

R: Changed as suggested. see lines 108-110. 

Comment 8-Line 128: These two species

R: Changed as suggested, see line 128.

Comment 9-Line 142: The landscape not only consist of...

R: Changed as suggested, see line 142.

Comment 10-Line 191: occurence records

R: Changed as suggested, see line 191. 

Comment 11-Lines 217: The model with the highest weight was the null model. I suggest the authors to rephrase this sentence so that it sound more clear and honest.

R: Changed as suggested. We have rewritten this sentence to make it clearer that the null model was the chosen one; see lines 216-218.

Comment 12-Line 235: was markedly

R: Changed as suggested, see line 238. 

Comment 13-Line 241-243: It is not clear whether P. cuvieri occur in areas where the maximum temperature is lower or higher thans its VTmax

R: We agree with the reviewer. We rephrased it to make it clearer, see line 245.

Comment 14-Lines 250-259: The results presentend in this part of the text should be summarized through charts. It was difficult to withdraw conclusions from Figure 2. I suggest that Figure 2 should be included in the Supplementary material and a chart (barplots or boxplots) should be presented along the manuscript instead.

R: We agree with the reviewer that the figure was not easy to interpret and have changed to barplots, as suggested. Thus, we also have moved the previous Figure 2 (map) to the Supplementary Material (S2 Fig).

Comment 15-Line 330: tend to be abundant in both open and non-open areas

R: Changed as suggested, see line 336.

---

## [Editor Report · Decision Letter 2]

8 Sep 2020

Does behavioral thermal tolerance predict distribution pattern and habitat use in two sympatric Neotropical frogs?

PONE-D-20-10531R2

Dear Dr. Diaz-Ricaurte,

We’re pleased to inform you that your manuscript has been judged scientifically suitable for publication and will be formally accepted for publication once it meets all outstanding technical requirements.

Kind regards,

Stefan Lötters

Academic Editor

PLOS ONE
---

## [Editor Report · Acceptance letter]

14 Sep 2020

PONE-D-20-10531R2 

Does behavioral thermal tolerance predict distribution pattern and habitat use in two sympatric Neotropical frogs? 

Dear Dr. Díaz-Ricaurte:

I'm pleased to inform you that your manuscript has been deemed suitable for publication in PLOS ONE. Congratulations! Your manuscript is now with our production department. 

Kind regards, 

on behalf of

Prof. Dr. Stefan Lötters 

Academic Editor

PLOS ONE